# Cellular Phosphorylation Signaling and Gene Expression in Drought Stress Responses: ABA-Dependent and ABA-Independent Regulatory Systems

**DOI:** 10.3390/plants10040756

**Published:** 2021-04-13

**Authors:** Fumiyuki Soma, Fuminori Takahashi, Kazuko Yamaguchi-Shinozaki, Kazuo Shinozaki

**Affiliations:** 1Laboratory of Plant Molecular Physiology, Graduate School of Agricultural and Life Sciences, The University of Tokyo, 1-1-1 Yayoi, Bunkyo-ku, Tokyo 113-8657, Japan; afsoma@mail.ecc.u-tokyo.ac.jp (F.S.); akys@g.ecc.u-tokyo.ac.jp (K.Y.-S.); 2Gene Discovery Research Group, RIKEN Center for Sustainable Resource Science, 3-1-1 Koyadai, Tsuku-ba, Ibaraki 305-0074, Japan; 3Department of Biological Science and Technology, Graduate School of Advanced Engineering, Tokyo University of Science, 6-3-1 Niijyuku, Katsushika, Tokyo 125-8585, Japan; 4Research Institute for Agricultural and Life Sciences, Tokyo University of Agriculture, Tokyo 156-8502, Japan

**Keywords:** drought, cellular signaling, gene expression, protein kinases, transcription factors, abscisic acid (ABA)

## Abstract

Drought is a severe and complex abiotic stress that negatively affects plant growth and crop yields. Numerous genes with various functions are induced in response to drought stress to acquire drought stress tolerance. The phytohormone abscisic acid (ABA) accumulates mainly in the leaves in response to drought stress and then activates subclass III SNF1-related protein kinases 2 (SnRK2s), which are key phosphoregulators of ABA signaling. ABA mediates a wide variety of gene expression processes through stress-responsive transcription factors, including ABA-RESPONSIVE ELEMENT BINDING PROTEINS (AREBs)/ABRE-BINDING FACTORS (ABFs) and several other transcription factors. Seed plants have another type of SnRK2s, ABA-unresponsive subclass I SnRK2s, that mediates the stability of gene expression through the mRNA decay pathway and plant growth under drought stress in an ABA-independent manner. Recent research has elucidated the upstream regulators of SnRK2s, RAF-like protein kinases, involved in early responses to drought stress. ABA-independent transcriptional regulatory systems and ABA-responsive regulation function in drought-responsive gene expression. DEHYDRATION RESPONSIVE ELEMENT (DRE) is an important cis-acting element in ABA-independent transcription, whereas ABA-RESPONSIVE ELEMENT (ABRE) cis-acting element functions in ABA-responsive transcription. In this review article, we summarize recent advances in research on cellular and molecular drought stress responses and focus on phosphorylation signaling and transcription networks in *Arabidopsis* and crops. We also highlight gene networks of transcriptional regulation through two major regulatory pathways, ABA-dependent and ABA-independent pathways, that ABA-responsive subclass III SnRK2s and ABA-unresponsive subclass I SnRK2s mediate, respectively. We also discuss crosstalk in these regulatory systems under drought stress.

## 1. Introduction

Drought stress is an environmental factor that results in reduced plant growth and yield productivity. Understanding the mechanisms that mediate drought stress responses and tolerance in plants can help improve crop productivity under drought stress [1,2,3,4]. When plants sense drought stress, rapid transcription of stress-related genes occurs through molecular signal transduction, such as protein phosphorylation, in the early cellular response to drought stress. The phytohormone abscisic acid (ABA) plays fundamental roles in plant responses to drought stress by regulating stomatal closure and gene expression [5,6]. In seed plants, there are three subclasses of SNF1-RELATED PROTEIN KINASES 2 (SnRK2s). Among them, subclass III SnRK2s act as key positive regulators of ABA signaling downstream of ABA receptors [7]. Subclass III SnRK2s phosphorylate and activate several transcription factors, such as ABRE-BINDING PROTEINS (AREBs)/ABRE-BINDING FACTORS (ABFs), to induce the expression of stress-responsive genes in an ABA-dependent manner [8]. In addition to these ABA-activated subclass III SnRK2s, seed plants have another type of ABA-unresponsive subclass I SnRK2s [9]. Subclass I SnRK2s regulate functional proteins, including mRNA decay factors, to adjust transcriptional stability and mediate plant growth under drought and salinity stress in an ABA-independent pathway [9,10,11,12]. DEHYDRATION-RESPONSIVE ELEMENT (DRE)-BINDING PROTEIN 2A (DREB2A) mediates transcriptional changes to acquire stress resistance in ABA-independent pathways [3,13].

In this review, we summarize recent knowledge of how plants regulate intracellular signal and gene expression through two different phosphorylation signaling pathways, ABA-dependent or ABA-independent pathways, in drought stress responses. We also discuss crosstalk of these regulatory systems in drought stress responses.

## 2. Importance of ABA-Dependent Responses under Drought Stress

The phosphorylation process was shown to play important roles in ABA signaling because ABA INSENSITIVE 1 (ABI1) was identified as a major protein phosphatase 2C in ABA signaling. In 2002, *Arabidopsis* SNF1-RELATED PROTEIN KINASES 2 (SnRK2s) was shown to play important roles in ABA-responsive stomatal closure and ABA-dependent gene expression [14,15,16]. One of the SnRK2s, SRK2E/SnRK2.6/OST1, is expressed in guard cells and is important for ABA-induced stomatal closure in *Arabidopsis*. SnRK2s are classified into 3 subclasses, I, II and III, and have been shown to function in not only ABA-responsive regulation but also ABA-independent regulation of drought stress responses [17,18]. Subclass III SnRK2s play key roles in ABA-dependent signaling pathways. Subclass II SnRK2s are also involved in ABA-dependent pathways [19]. However, subclass I SnRK2s are not activated by ABA. Here, we focus on the key roles of subclass III SnRK2s and their upstream and downstream factors in ABA-dependent cellular signaling.

### 2.1. ABA-Dependent Phosphorylation Signaling in the Dehydration Stress Response: Important Roles of Subclass III SnRK2 Protein Kinases

In *Arabidopsis*, three subclass III SnRK2s, SRK2D/SnRK2.2, SRK2E/SnRK2.6/OST1 and SRK2I/SnRK2.3, are mainly involved in ABA-dependent drought stress signaling. The *srk2dei* (*snrk2.2/snrk2.6/snrk2.3*) triple mutant lacking all subclass III SnRK2s shows severe ABA-insensitive viviparity, ABA-insensitive germination and a drought-sensitive phenotype [20,21,22,23]. These observations show the key roles of subclass III SnRK2s in ABA responses in diverse developmental regulatory processes, such as germination, seed maturation and drought tolerance. The PYRABACTIN RESISTANCE1/PYR1-LIKE/REGULATORY COMPONENTS OF ABA RECEPTOR (PYR/PYL/RCAR) family contains important ABA receptors that bind to ABA to regulate ABA responses [20,23,24,25,26]. Their role in ABA perception is established as shown below (Figure 1). In the absence of ABA, the kinase activity of subclass III SnRK2s is inhibited by group A PROTEIN PHOSPHATASES 2C (PP2Cs) [17]. When plants are subjected to drought stress, ABA accumulates in plant cells in large amounts, and then the ABA receptors PYR/PYL/RCAR bind to ABA. ABA-PYR/PYL/RCAR complexes competitively interact with PP2C and release SnRK2s [20,23,24,25,26]. The released SnRK2s are activated by autophosphorylation or phosphorylation by other protein kinases [17,27] (Figure 1). These three components, PYR/PYL/RCAR, PP2C and SnRK2, are now well known as core components in ABA sensing and signaling.

One of the phosphorylation targets of subclass III SnRK2s is the group A bZIP transcription factor ABA-RESPONSIVE ELEMENT BINDING PROTEIN/ABA-RESPONSIVE ELEMENT BINDING FACTOR (AREB/ABF), which regulates the expression of ABA-inducible genes [28,29] in response to drought stress or ABA [30,31]. AREB/ABF is activated through multisite phosphorylation of conserved domains by SnRK2s [31]. In particular, AREB1/ABF2, AREB2/ABF4, ABF3 and ABF1 redundantly regulate stress-inducible gene expression [5,32]. These ABA-SnRK2-AREB/ABF signaling modules can be formed in vitro or in yeast cells [20,33].

Phosphoproteomics studies using *srk2dei* triple mutants lacking all subclass III SnRK2s have been performed to investigate SnRK2-mediated phosphorylation signaling [34,35]. MITOGEN-ACTIVATED PROTEIN KINASE (MPK) 1, MPK2, SnRK2 SUBSTRATE 1 (SNS1) and bZIP transcription factors, including AREB1/ABF2, AREB3 and EEL, were identified as candidate substrates of subclass III SnRK2s by these analyses [34,35]. SNS1 is an unknown protein, but an *sns1* mutant showed an ABA-hypersensitive phenotype in the post-germination stage [34]. Although these factors may be involved in SnRK2-mediated phosphorylation signaling, further analyses are required to reveal whether these factors are directly or indirectly regulated by SnRK2s. Phosphoproteomic analyses also revealed that the subclass III SnRK2-AREB/ABF phosphorylation signaling module is well conserved in *Oryza sativa* [36], *Brachypodium distachyon* [37], and *Solanum lycopersicum* [38]. Recently, a new approach was reported to integrate in vivo and in vitro phosphoproteomic information to show the direct targets of protein kinases [39]. Extensive analyses of *Arabidopsis* proteins were performed using genome-wide phosphoproteomics data from plant cells, which is an important basis for phosphorylation signaling in *Arabidopsis* [40].

### 2.2. Regulatory Mechanisms of Subclass III SnRK2s

Subclass III SnRK2s are activated in ABA-deficient and ABA-insensitive mutants, suggesting that SnRK2s can be activated by upstream kinases in an ABA-independent manner under drought stress [17,27,41,42]. B-type MPK KINASE KINASE (MAPKKK) Raf-like kinases have been shown to directly regulate the activity of SnRK2s under osmotic stress [12,43,44,45]. An ABA-insensitive *Physcomitrella patens* mutant was isolated by a forward genetic approach and was shown to have a mutation in a gene for ABA AND ABIOTIC STRESS-RESPONSIVE RAF-LIKE KINASE/ABA NON-RESPONSIVE (ARK/ANR), one of the B3-MAPKKK Raf-like kinases [46,47]. This gene was previously identified as a PpCTR1 that regulates ABA and ethylene responses in *Physcomitrella patens* [48]. The ARK/ANR kinase directly phosphorylates and activates SnRK2s [46,49]. Three ARK/ANR homologs, M3Kδ1/RAF3, M3Kδ6/RAF5/SIS8 and M3Kδ7/RAF4, which are classified as B3 Raf-like kinases, were identified as activators of subclass III SnRK2s in *Arabidopsis* [50]. RAF6, which is the closest homolog of CTR1/RAF1 and a member of B3 Raf-like kinases, also regulates subclass III SnRK2s under osmotic stresses [50,51]. RAF10, one of the B2 Raf-like kinases, which are the closest family members of B3 Raf-like kinases, can also phosphorylate and activate subclass III SnRK2s by releasing them from PP2Cs [52,53] (Figure 1). These reports suggest that B2 Raf-like kinases may also be involved in the activation of subclass III SnRK2s. Recently, three B4 Raf-like kinases, RAF18, RAF20 and RAF24, were identified as activators of ABA-unresponsive subclass I SnRK2s under osmotic stress [54]. In addition, the activities of SnRK2s were significantly decreased in a multiple mutant lacking B2, B3 and B4 Raf-like kinases under osmotic stress [55]. It is necessary to elucidate different roles between B2 and B3 Raf-like kinases to reveal the physiological and functional roles of subclass III SnRK2 activation by Raf-like kinases.

In maize, CASEIN KINASES 2 (CK2s) phosphorylate SnRK2s, increasing their binding to PP2Cs [56]. However, CK2-knockout mutants show reduced ABA sensitivity [57]. CK2 catalytic subunits (CK2αs) positively regulate ABA-dependent gene expression, but the CK2 regulatory subunit (CK2β) negatively regulates ABA-dependent gene expression [58]. These findings suggest that the balance of CK2 subunits determines ABA responses (Figure 1). Subclass III SnRK2s are also phosphorylated by BRASSINOSTEROID INSENSITIVE 2 (BIN2), a GLYCOGEN SYNTHASE KINASE 3 (GSK3)-like kinase [59]. BIN2 interacts with and phosphorylates subclass III SnRK2s to promote their activity [59] (Figure 1). Taken together, crosstalk between ABA and brassinosteroids may occur under stress conditions via subclass III SnRK2s.

The activity of subclass III SnRK2s is also regulated by their stability. AtPP2-B11 and HOS15, which are components of E3 ligase complexes, promote the degradation of SnRK2s in the presence of ABA [60,61]. SnRK2 INTERACTING CALCIUM SENSOR (SCS) inhibits the activity of SnRK2s in the presence of Ca^2+^ in plant cells [62,63]. As the cytosolic concentration of Ca^2+^ is significantly increased in response to osmotic stress, SCS may regulate the activity of SnRK2s in the later stage of osmotic stress signaling.

### 2.3. Other Regulatory Mechanisms of ABA Receptors and Coreceptors

The activities of subclass III SnRK2s are also controlled by mediators other than ABA receptors. Degradation of PP2Cs is an additional regulatory mechanism for the inhibition of PYR/PYL/RCAR receptor activities. Among clade A, PP2Cs, PP2C, ABI1, ABI2, PP2CA and HAB1 are essential for osmotic stress responses [64]. ABI1 is degraded by two U-box E3 ligases, PUB12 and PUB13, and this degradation is promoted by ABA [65]. On the other hand, PP2CA is degraded by several protease systems, such as the E3 ubiquitin ligases RGLG1, RGLG5 [66], PIR1.1, and PIR1.2 [67] and the BPM-CUL3 E3 ligases BPM3 and BPM5 [68], to enhance ABA responses. DELAY OF GERMINATION 1 (DOG1), which encodes a protein of unknown function, suppresses ABA-HYPERSENSITIVE GERMINATION (AHG) 1/3 encoding a clade A PP2C [69,70] in the germination stage. Because DOG1 positively regulates drought tolerance [71], DOG1-mediated PP2C suppression may function in the drought stress response. These findings suggest that PP2Cs are regulated by several mechanisms other than PYR/PYL/RCAR-dependent regulation by ABA to promote survival under changing environmental conditions by precisely controlling ABA responses.

Regulation of ABA receptors is also involved in ABA signaling under drought stress. CARK1, a putative receptor-like cytoplasmic kinase, phosphorylates PYL8/RCAR3 and PYR1/RCAR11 to promote its stability and ability to inhibit ABI1, leading to the activation of ABA signaling in the presence of ABA [72]. On the other hand, *Arabidopsis* EL1-LIKE (AEL) phosphorylates PYR/PYL/RCAR to decrease its stability, leading to suppression of ABA responses under non-stress conditions [73]. These findings suggest that ABA signaling is precisely regulated by controlling the stability of PYR/PYL/RCAR.

Dehydration stress-resistant crops can be produced by overexpression or mutation of ABA receptors. ABA sensitivity and drought tolerance are significantly improved in PYL-overexpressing wheat plants (TaPYLox) compared with wild-type plants [74]. Biomass production and seed production are also improved by ABA receptor overexpression because of the reduction in transpiration and enhancement of photosynthetic activity in wheat [74]. ABA signaling suppresses plant growth under non-stress conditions [18,75]. Rice growth and productivity can be improved by CRISPR/Cas9-mediated mutations in rice PYR/PYL/RCAR [76]. Several combinations of PYL mutations have been constructed and analyzed in rice. Among them, *pyl1/4/6* exhibited the best growth and improved grain productivity under field conditions [76]. These methods provide novel insight into approaches to improve drought stress tolerance and plant growth in crops by applying the factors involved in ABA responses in *Arabidopsis*.

## 3. ABA-Independent Signaling in Response to Dehydration Stress: Important Roles of Subclass I SnRK2s

In drought stress responses, ABA-independent regulatory systems have been identified as playing important roles in addition to ABA-dependent systems. Under drought stress, plants sense and respond to water deficit signals in the roots; then, the stress signal is transported to aerial parts, including the leaves, to synthesize ABA and close stomata to reduce water loss. ABA also induces many stress genes with various functions to cope with dehydration stress. In the early processes of drought stress responses before the accumulation of endogenous ABA, ABA-independent processes function in sensing and signaling to activate regulatory systems to respond to water deficit stress, including osmotic stress, ROS stress, mechanical stress and other stress signals. Then, these early stress signals upregulate ABA biosynthesis by the induction of the *9-CIS-EPOXYCAROTENOID DIOXYGENASE 3* (*NCED3*) gene [6]. Therefore, for a proper understanding of plant responses to drought stress, the early processes of the ABA-independent regulatory system must be included. In cellular signal transduction, subclass I SnRK2 protein kinases play important roles in ABA-independent signaling pathways. In this section, we discuss the phosphorylation processes and focus on the roles of subclass I SnRK2 protein kinases.

### 3.1. Subclass I SnRK2-Mediated Dehydration Stress Signaling

Four of the ten SnRK2s, SRK2A/SnRK2.6, SRK2B/SnRK2.10, SRK2.1/SnRK2.10 and SRK2H/SnRK2.5, which are classified as subclass I SnRK2s in *Arabidopsis*, are strongly activated in response to osmotic stress but not ABA [17]. The kinase activities of subclass I SnRK2s are enhanced by dehydration stress prior to ABA accumulation and in ABA-deficient plants [11,17]. Although subclass I SnRK2s are involved in the osmotic stress response and cadmium response, the substrates of these kinases have not been elucidated [11,77]. Interactome analyses have revealed that subclass I SnRK2s interact with VARICOSE (VCS), which is a scaffold protein of mRNA-decapping complexes [9]. The expression of many stress-repressive genes was similarly upregulated in *srk2abgh,* a mutant lacking all functional subclass I SnRK2s, and in *VCS*-knockdown plants under osmotic stress. Additionally, mRNA decay of the transcripts of these genes was impaired in these plants under osmotic stress. On the other hand, the expression of many stress-induced genes decreased in *srk2abgh* and VCS-knockdown plants compared with wild-type plants under osmotic stress. These observations suggest that the expression of drought stress-responsive genes is positively regulated by posttranslational regulation mediated by the subclass I SnRK2-VCS signaling module [9] (Figure 1). The subclass I SnRK2s-VCS signaling module is also involved in modifying root architecture and development in response to salt stress [10]. DCP1, one of the mRNA decapping activators, is also phosphorylated by MPK6 in response to dehydration stress [78]. These reports suggest that phosphorylation of decapping complexes is important for the activation of mRNA decapping and decay. Phosphoproteomic analyses revealed that SRK2B/SnRK2.10 phosphorylates two dehydrins, EARLY RESPONSIVE TO DEHYDRATION (ERD)10 and ERD14 [79]. Phosphorylation of ERDs is involved in the regulation of subcellular localization to acquire drought tolerance [79] (Figure 1). Subclass I SnRK2s also regulate the expression of several genes responsible for ROS generation and removal under salt stress [80].

### 3.2. Regulatory Components of Subclass I SnRK2s

All functional subclass I SnRK2s are phosphorylated in the early stage of osmotic stress [81]. The Ser-154 residue of subclass I SnRK2s is phosphorylated by unknown kinases in response to osmotic stress [27,42]. Recently, three B4 Raf-like kinases, RAF18, RAF20 and RAF24, were identified as protein kinases responsible for phosphorylating Ser-154 of subclass I SnRK2s and activating them in response to osmotic stress [54]. The activities of subclass I SnRK2s under osmotic stress were largely decreased in a *raf18/20/24* triple mutant [54]. Considering that the three Raf-like kinases and subclass I SnRK2s are well conserved in seed plants but not in moss [9,54], the RAF18/20/24-subclass I SnRK2-VCS signaling module may have evolved in seed plants to control mRNA populations and to enhance adaptability to stress conditions (Figure 1). Another report of a genetic approach using a septuple mutant lacking all B4 Raf-like kinases also suggested that B4 Raf-like kinase is responsible for the activation of subclass I SnRK2s [55]. The growth retardation of the septuple mutant was more severe than that of the *raf18/20/24* or *srk2abgh* mutant under osmotic stress, indicating the existence of other substrates of B4 Raf-like kinases. Further analyses are required to reveal the other substrates of B4 Raf-like kinases.

The activity of subclass I SnRK2s is also controlled by other mediators in addition to RAF18/20/24 kinases. The activities of SRK2A/SnRK2.4 and SRK2B/SnRK2.10 are regulated by ABI1 and PP2CA under osmotic stress [82]. Because of the existence of a phosphatidic acid (PA)-binding domain in subclass I SnRK2s [83], PA is thought to regulate the kinase activity of these kinases (Figure 1). As osmotic stress-induced accumulation of PA in roots is dependent on PHOSPHOLIPASE Dα (PLDα1) [84], the activity of SnRK2s might be regulated by PLDα1.

In summary, the importance of ABA-activated and ABA-independent SnRK2s in dehydration stress responses has been discussed (Figure 1). ABA-activated subclass III SnRK2s regulate the expression of many stress-responsive genes and stomatal responses under dehydration stress. On the other hand, ABA-independent and dehydration-activated subclass I SnRK2s regulate the mRNA population to correctly control stress-induced gene expression. However, the osmosensors that receive an osmotic signal and induce ABA accumulation or directly activate RAF-SnRK2 in an ABA-independent manner are largely unknown. There are many reports on candidates for sensing molecules of dehydration stress [85,86]. However, major sensory systems have not yet been identified. In the next step, it is important to analyze upstream factors of Raf1 and subclass I SnRK2 to identify sensors of dehydration stress signals.

## 4. Transcriptional Regulation of Cellular Signaling in Drought Stress Responses

Various types of transcription factors mediate the regulation of gene expression in response to drought stress. Drought stress-responsive genes play crucial roles in the drought stress tolerance of plants [8,87]. Higher plants have more transcription factors than animals do to adapt to environmental stresses, thereby indicating that the networks for transcriptional regulation in plants are more complicated than those in animals, which can move to reduce or avoid environmental stresses. In plants, ABA plays important roles in drought stress-responsive gene expression. ABA-independent transcriptional regulatory systems also function in dehydration stress responses in addition to ABA-dependent systems [87]. In this section, we summarize the functions of transcription factors that mediate drought, heat and ABA responses in plants.

### 4.1. Transcription Factors That Mediate ABA-Responsive Gene Expression

Many drought stress-responsive genes have cis-acting ABREs (PyACGTGGC) in their promoter regions [87,88]. The ABA-RESPONSIVE ELEMENT (ABRE) motif is mainly recognized by bZIP-type transcription factors, AREBs/ABFs (Figure 2). There are nine subfamily members of AREBs/ABF in *Arabidopsis*. Among them, AREB1/ABF2, AREB2/ABF4 and ABF3 have conserved amino acid sequences that are phosphorylated by subclass III SnRK2s (SRK2D/SnRK2.2, SRK2E/SnRK2.6 and SRK2I/SnRK2.3) and mediate the expression of drought- and ABA-responsive genes through binding to the ABRE motif in their promoter region [21,89]. The substitution of phosphorylated amino acids with aspartic acid led to the enhancement of AREB1 transcriptional activity even without ABA treatment, indicating that phosphorylation modification is required for the regulation of transcriptional activity in AREBs/ABF under stress conditions [31]. Triple mutants of *AREB1*, *AREB2* and *ABF3* showed a reduction in drought- and ABA-responsive gene expression and drought stress-sensitive phenotypes [90]. These results indicate that AREB1/ABF2, AREB2/ABF4 and ABF3 are key transcription factors that regulate drought stress responses through an ABA-dependent pathway. Remarkably, COUPLING ELEMENT 3 (CE3) is often located in the vicinity of ABREs on the promoters of ABA-responsive genes [91]. Further analysis will reveal the functional relationship between CE3 and ABRE that regulates the binding of transcription factors under drought stress.

### 4.2. Transcription Factors for Drought- and Heat-Responsive Gene Expression in ABA-Independent Signaling

The DREB2A transcription factor is expressed under dehydration and osmotic stress [92]. DREB2A binds to the DRE motif (A/GCCGAC) present in the promoter regions of drought stress-responsive genes [88] (Figure 2). Domain analysis of DREB2A revealed that a negative regulatory domain (NRD) located in the central region of DREB2A is important for the regulation of DREB2A transcriptional activity [93]. Transgenic plants overexpressing constitutively active forms of DREB2A (DREB2A CA) that lack the NRD show enhancement of both stress-inducible gene expression and dehydration stress resistance. The abiotic stress-induced overexpression of DREB2A CA improved the phenotype of drought stress resistance in transgenic *Arabidopsis* and soybean without growth defects [93,94]. The stability of the DREB2A protein is regulated by modification of the NRD. DREB2A-INTERACTING PROTEIN 1 (DRIP1) and DRIP2 were isolated as E3 ubiquitin ligases that mediate the ubiquitination of DREB2A [95]. Ubiquitinated DREB2A is degraded through the 26S proteasome pathway under control conditions.

Drought stress responses are known as complex physiological responses that include not only dehydration stress responses but also heat stress responses. Other research indicated that overexpression of DREB2A CA in plants mediated the enhancement of heat-induced gene expression and resulted in thermotolerance [96]. The stability of DREB2A is regulated by other degradation pathways under heat stress. BTB/POZ AND MATH DOMAIN PROTEINS (BPMs) encode substrate adaptors of the CULLIN3 (CUL3)-based E3 ligase and bind to the NRD of DREB2A [97]. *BPM*-knockdown plants accumulated the DREB2A protein and showed enhanced expression of DREB2A target genes under heat stress and improved heat stress tolerance. DNA POLYMERASE II SUBUNIT B3-1 (DBP3-1)/NUCLEAR FACTOR-Y SUBUNIT C10 (NF-YC10), NF-YA and NY-YB interact with DREB2A and enhance the transcriptional activity of DREB2A under heat stress [98]. The inhibition of phosphorylation in the NRD through CASEIN KINASE 1 (CK1) stabilizes and activates DREB2A in response to heat stress [99]. These results indicate that DREB2A mediates drought and heat stress signaling via the different regulatory mechanisms of the NRD in DREB2A.

### 4.3. Crosstalk in Transcriptional Regulation between Drought and Other Stress Responses

Cellular responses to drought stress are regulated through various types of transcription factors [8]. AREB/ABFs partially mediate the expression of *DREB2A* under dehydration stress [100], suggesting crosstalk between early processes of dehydration stress before the accumulation of ABA and slow adaptive ABA-dependent processes of stress-responsive gene expression. Moreover, various types of transcription factors have been reported to function in dehydration-responsive transcription. Most of them function mainly in crosstalk between drought and other stress responses, such as wounding, active oxygen, pathogen infection and other environmental stress signals. In this section, we focus on HOMEODOMAIN-LEUCINE ZIPPER (HD-Zip), NAM, ATAF1, 2 and CUC2 (NAC) transcription factors in the crosstalk between drought and other stress responses.

An HD-Zip transcription factor, HOMEOBOX PROTEINS 6 (HB6), negatively regulates ABA responses [101]. HB6 interacts with ABA INSENSITIVE 1 (ABI1) and binds to the core motif (CAATTATTA) that ABA-responsive genes have in their promoter regions. Transgenic plants overexpressing *HB6* show an ABA-insensitive phenotype during seed germination. HB6 inhibits stomatal closure in detached leaves. These observations support the role of HB6 mainly in the crosstalk between ABA responses under drought stress.

The NAM, ATAF1, 2 and CUC2 (NAC) genes are plant-specific transcription factors that bind to the consensus NAC recognition site (NACR, CGTG/A) [102] (Figure 2). Previous studies indicate that a subfamily of stress-responsive NAC transcription factors (SNACs) mediate environmental stress responses [103,104]. ANAC096 is expressed in response to drought stress and enhances the expression of drought-induced genes. Specifically, ANAC096 mediates the expression of the *RESPONSIVE-TO-DEHYDRATION 29A* (*RD29A*) gene, which is one of the marker genes in drought stress signaling, through interaction with AREB1/ABF2 and AREB2/ABF4. *anac096* mutants show a dehydration-sensitive phenotype [105]. These results indicate that ANAC096 functions as a positive regulator in drought stress signaling. On the other hand, ANAC016 negatively regulates the expression of drought stress-induced genes, including *AREB1*/*ABF2*, although the expression of the *ANAC016* gene is enhanced in response to drought stress [106], which may cause the drought stress-resistant phenotype of *anac016* mutants. Detailed phylogenetic analysis has revealed that SNACs are divided into two major branches, SNAC-A and SNAC-B. Seven SNACs, ANAC002/ATAF1, ANAC019, ANAC032, ANAC055, ANAC072/RD26, ANAC081/ATAF2 and ANAC102, were included in the SNAC-A group and expressed under drought and ABA treatment. These seven SNACs mediate ABA-dependent leaf senescence under drought stress [107]. The septuple mutant of SNACs showed a delayed leaf senescence phenotype in response to ABA treatment. Comparative transcriptome analysis revealed that major senescence-related genes were repressed in the SNAC septuple mutants.

## 5. ABA Crosstalk with Other Hormones under Drought Stress

Transcriptional regulation via transcription factors mediates the crosstalk between ABA and other hormone signaling pathways. Among them, the regulatory roles of MYC, MYB, WRKY and ERF transcription factors are discussed in this section. AtMYC2, encoding a basic helix loop helix (bHLH) transcription factor, is induced by ABA and regulates ABA-inducible transcription of the drought-inducible *RD22* gene and *AtADH1* gene [108,109]. AtMYC2 is also a master regulator of jasmonate (JA) signaling pathways and controls many JA-dependent processes, from insect and pathogen defense to plant development [110]. A unique function of MYC2 is its regulatory role in the crosstalk between hormone signaling pathways involving JA and those involving ABA, salicylic acid (SA), gibberellin (GA) and auxin (IAA) [110,111]. AtMYB2, encoding an R2R3-type MYB-related transcription factor, regulates RD22 together with AtMYC2 [109], which indicates that the interaction between AtMYC2 and MYB2 determines their functions in ABA-inducible transcription. Another MYB transcription factor, MYB96, mediates dehydration stress responses through the regulation of crosstalk between ABA and auxin signals [112]. MYB96 is expressed not only in the vasculature of leaves and roots but also in stomata in response to ABA and dehydration stress. Transgenic plants overexpressing MYB96 show reduced shoot growth and fewer lateral roots, thereby enhancing dehydration stress resistance.

The WRKY transcription factors that bind to the W-box (TTGACC/T) mediate various abiotic and biotic stress responses and act as activators or repressors of particular physiological responses [113] (Figure 2). Previous studies indicate that WRKY18, WRKY40 and WRKY60, which belong to the group IIa WRKY family, are involved in ABA signaling [114]. WRKY40 directly binds to the promoters of the *AREB2*/*ABF4*, *ABI4*, *ABI5*, *MYB2*, *DREB1A* and *RAB18* genes under control conditions [115,116]. This suppressive effect of WRKY40 is eliminated in response to ABA treatment. Triple knockout mutants of WRKY18, WRKY40 and WRKY60 showed an ABA-hypersensitive phenotype, such as growth arrest during ABA-mediated seed germination prior to the post-germination stage. These results indicate that WRKY18, WRKY40 and WRKY60 act as repressors in ABA responses. ABA overly sensitive 3 (ABO3)/WRKY63 was identified as the causative gene of both root growth inhibition and germination inhibition in ABA hypersensitive mutants [117]. *abo3* mutants showed impaired ABA-induced stomatal closure, thereby showing a dehydration-sensitive phenotype. ABO3/WRKY63 is able to bind the W-box in the *AREB1*/*ABF2* promoter. The expression of the *AREB1*/*ABF2*, *RD29A* and *COR47* genes was partially repressed in the *abo3* mutants under ABA treatment. On the other hand, *areb1*/*abf2* mutants show the opposite phenotype in ABA-mediated germination inhibition compared to that of *abo3* mutants. AREB1/ABF2 does not mediate ABA-induced stomatal closure. These results indicate that ABO3/WRKY63 mediates not only AREB1/ABF2-mediated signaling but also other transcriptional cascades in ABA and dehydration stress responses.

ETHYLENE RESPONSE FACTOR1 (ERF1) is a transcription factor that mediates pathogen responses through ethylene and jasmonate signaling. ERF1 is also able to bind to the DRE motif in the promoter regions of genes downstream of ERF1. ERF1 regulates stomatal closure to prevent water loss from guard cells. ERF1-overexpressing plants show a dehydration resistance phenotype. These results indicate that ERF1 activates specific sets of stress genes and functions in crosstalk between biotic and abiotic stress responses.

## 6. Conclusions and Future Perspectives

Plants have a sophisticated stress response mechanism that provides a balance between stress tolerance and optimal growth under drought stress. The phytohormone ABA mediates various physiological responses under dehydration stress, including stomatal closure in guard cells and stress-mediated gene expression in various plant tissues. ABA strongly accumulates in response to drought stress through peptide signaling, phosphorylation signaling and gene expression. In addition to an ABA-dependent regulatory system in dehydration responses, an ABA-independent regulatory system including signal transduction and transcription also plays important roles in early processes of the stress response before the accumulation of ABA. In an ABA-dependent pathway, subclass III SnRK2s and AREB/ABF are the main positive regulators of dehydration stress responses. ABA-activated subclass III SnRK2s mediate the expression of stress-responsive genes and enhance stress tolerance through the phosphorylation of AREB/ABF transcription factors. In addition to the subclass III SnRK2-AREB/ABF pathway, plants have evolved a subclass I SnRK2-VCS pathway. The subclass I SnRK2-VCS pathway regulates changes in the transcriptome by activating mRNA decay under dehydration stress via an ABA-independent pathway. The fine tuning of transcriptomic changes is required for optimal growth without growth retardation under drought stress because some stress-mediated genes adversely affect both plant growth and yield.

Recent phosphoproteomics analyses revealed that there are many regulators of subclass III SnRK2s, ABA receptors or coreceptors that mediate the regulation of gene expression through the AREB/ABF transcriptional pathway. Among them, plant Raf-like kinases have been newly identified as upstream factors of SnRK2s. Considering that Raf-like kinases phosphorylate and activate almost all subclass III SnRK2s under osmotic stress, we need to redefine the physiological relationship between group A PP2Cs and Raf-like kinases with regard to SnRK2 activation.

Furthermore, an ABA-independent regulatory system functions in early processes of dehydration stress responses before the accumulation of ABA. In the early processes of dehydration responses, the identification of osmosensors that may directly regulate the activity of Raf-like kinases will provide new insights into how phosphorylation signaling pathways are activated quickly by dehydration stress. Further studies will also reveal the downstream regulation of SnRK2s through transcription factors such as AREB/ABF, DREB2A and other transcription factors in dehydration responses. A comprehensive understanding of regulatory mechanisms through gene expression will contribute to the discovery of useful genes for the improvement of yield productivity and optimal growth under drought stress.

## Figures and Tables

**Figure 1 plants-10-00756-f001:**
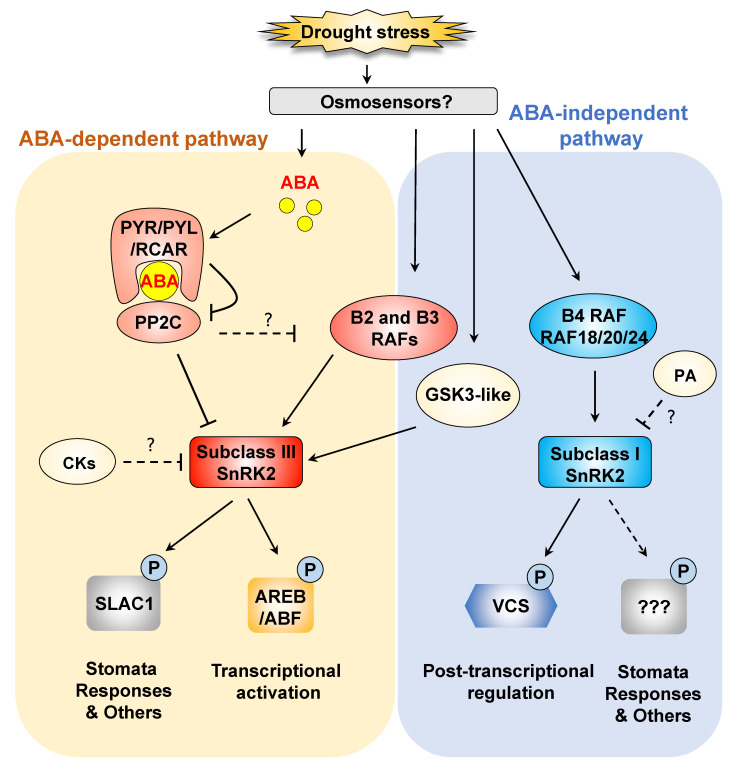
SnRK2-mediated drought stress signaling and gene expression. Subclass III SNF1-related protein kinases 2 (SnRK2s) play key roles in abscisic acid (ABA) signaling under osmotic stress. Negative regulation of subclass III SnRK2s is released via inhibition of protein phosphatases 2C (PP2C) by ABA-pyrabactin resistance1/pyr1-like/regulatory components of ABA receptor (ABA-PYR/PYL/RCAR) complexes under osmotic stress. Subclass III SnRK2s are phosphorylated and activated by B-type Raf-like kinases and GSK3-like kinases in an ABA-independent manner. Activated subclass III SnRK2s phosphorylate a variety of substrates, including ABA-responsive element binding proteins/ABA-responsive element binding factor (AREB/ABF), to induce the expression of stress-responsive genes. ABA-unresponsive subclass I SnRK2s are quickly activated in response to osmotic stress in an ABA-independent manner. Osmotic stress-activated subclass I SnRK2s regulate mRNA decay by phosphorylating the mRNA-decapping activator VCS. The B4 Raf-like kinases RAF18, RAF20 and RAF24 are responsible for the activation of subclass I SnRK2s. Dashed lines indicate possible but unconfirmed routes.

**Figure 2 plants-10-00756-f002:**
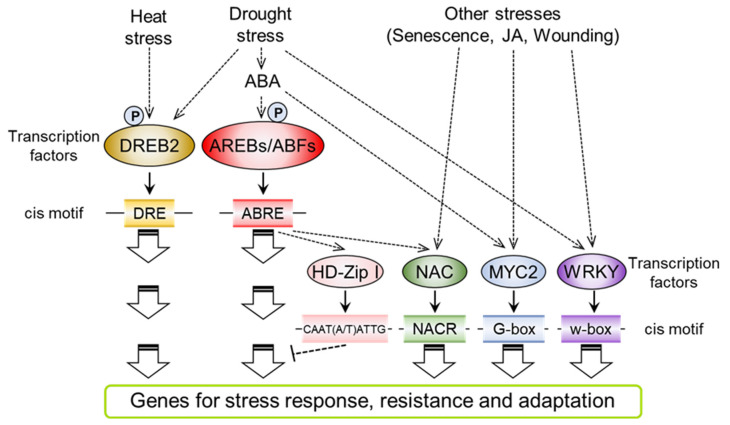
A schematic model of transcriptional regulatory networks under abiotic stress, including drought, heat and other stresses. The ellipsoids show transcription factors that mainly mediate drought stress and/or ABA responses. The oblong boxes indicate the cis motifs that exist in the promoter region of stress-induced genes. ABREs/ABFs function mainly in drought- and ABA-dependent pathways. DREB2 mediates ABA-independent and heat stress signals. Other transcription factors, including NAC, MYC2 and WRKY, mediate crosstalk signals between ABA and other stresses in plants..

## Data Availability

Not applicable.

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
