# Peer review of "Cellular Phosphorylation Signaling and Gene Expression in Drought Stress Responses: ABA-Dependent and ABA-Independent Regulatory Systems"

_plants, 2021, doi:10.3390/plants10040756_

Round 1

Reviewer 1 Report

The review from Soma et al. summarizes recent findings on drought stress signaling differentiating between ABA-dependent and ABA-independent pathways.

Overall, the review is very comprehensive and covers the topic well. I can follow the basic structure, but some parts are difficult to understand, some parts are redundant and the same results are presented in different paragraphs. Furthermore, the text contains a large number of small mistakes. Furthermore, the text rather heavily relies on publications of the lab of the corresponding authors.

Unfortunately, the text does not have any line-numbers so that it is difficult to point out the mistakes I found and I can only give general advice: please check if the time you use is correct, sometimes you jump between past tense and present tense where it is not adequate, check if you put all 3rd person singular "s" and vice versa, in several places articles are either missing or not necessary, some prepositions are missing or in the wrong place, please use gene expression (not expressions). There are many more grammatical mistakes, and the text should be polished by a native speaker.

In the following I will only point out the paragraphs that need rewriting/rearrangement:

I recommend to shorten the abstract and to focus it - it already provides too many details and is on the other hand redundant. I suggest to delete the sentences starting with Dehydration Responsive Element until dehydration response.

The last paragraph of the introduction is important but the first and second sentence are redundant.

You describe the three classes of SnRK2s in the introductory paragraph of 2. and at the beginning of 2.1. Please delete redundancy. The last sentence on page 2 and the first sentence on page 4 and the last sentence of the first paragraph on page 4 repeat the same message.

Please delete redundancy in the last sentences (starting with phosphoproteomic analyses) of the first paragraph on page 4.

The ABA-PYR/PYL/RCAR complex is mentioned and described on page 3, page 4 and page 5 - please streamline and subchapter 2.3 for redundancy.

I suggest to reframe the second sentence of subchapter 4: genes play a crucial role in drought stress resistance of plants.

On page 10 instead of overexpression plants of  please use plants overexpressing DREB2A....

Author Response

The review from Soma et al. summarizes recent findings on drought stress signaling differentiating between ABA-dependent and ABA-independent pathways.

Overall, the review is very comprehensive and covers the topic well. I can follow the basic structure, but some parts are difficult to understand, some parts are redundant and the same results are presented in different paragraphs. Furthermore, the text contains a large number of small mistakes. Furthermore, the text rather heavily relies on publications of the lab of the corresponding authors.

Unfortunately, the text does not have any line-numbers so that it is difficult to point out the mistakes I found and I can only give general advice: please check if the time you use is correct, sometimes you jump between past tense and present tense where it is not adequate, check if you put all 3rd person singular "s" and vice versa, in several places articles are either missing or not necessary, some prepositions are missing or in the wrong place, please use gene expression (not expressions). There are many more grammatical mistakes, and the text should be polished by a native speaker.

>ANSWER

Thank you very much for your constructive and kind suggestions. We submitted our manuscript with line numbers in the word file although the reviewers received a file that the format of the manuscript was changed to publication format. We proofed our revised manuscript with an English editing service. We revised several redundant sentences in accordance with the reviewer’s comments as below.

In the following I will only point out the paragraphs that need rewriting/rearrangement:

I recommend to shorten the abstract and to focus it - it already provides too many details and is on the other hand redundant. I suggest to delete the sentences starting with Dehydration Responsive Element until dehydration response.

>ANSWER

We deleted the sentences pointed out and shortened the abstract.

The last paragraph of the introduction is important but the first and second sentence are redundant.

>ANSWER

We summarized the first and second sentences to eliminate duplication in the last paragraph of the introduction.

You describe the three classes of SnRK2s in the introductory paragraph of 2. and at the beginning of 2.1. Please delete redundancy. The last sentence on page 2 and the first sentence on page 4 and the last sentence of the first paragraph on page 4 repeat the same message.

>ANSWER

We deleted and summarized the descriptions of ABA and SnRK2s in the introductory paragraph of 3 and the beginning of 3.1. In addition, we deleted and summarized the explanations of SnRK2 functions in ABA signaling in the paragraph of 3.1.

Please delete redundancy in the last sentences (starting with phosphoproteomic analyses) of the first paragraph on page 4.

>ANSWER

We deleted the redundant explanations of phosphoproteomics analyses in the third paragraph of the subchapter 3.1.

The ABA-PYR/PYL/RCAR complex is mentioned and described on page 3, page 4 and page 5 - please streamline and subchapter 2.3 for redundancy.

>ANSWER

We deleted and summarized the redundant explanations of ABA-PYR/PYL/RCAR complex among the first paragraph of the subchapter 3.1, the first paragraph of the subchapter 3.2 and the second paragraph of the subchapter 3.3.

I suggest to reframe the second sentence of subchapter 4: genes play a crucial role in drought stress resistance of plants.

>ANSWER

We revised the sentence pointed out in accordance with the suggestion.

On page 10 instead of overexpression plants of please use plants overexpressing DREB2A....

>ANSWER

We corrected the text according to the suggestions of the reviewer and the English proof editor.

Reviewer 2 Report

This manuscript provides a review of our current understanding of the roles of regulatory mechanisms involving phosphorylation and gene expression in responses to drought stress. The authors delineate both ABA-dependent and ABA-independent regulatory systems and demonstrate crosstalk between these systems.

The review is comprehensive of current developments as well as providing sufficient background of the past twenty years of research in this field. The authors are well established and leading researcher in this field.

Apart from some suggestions for some minor changes to English expression, the manuscript is most suitable for publication in Plants. I have included some suggested changes in the attached document.

Author Response

This manuscript provides a review of our current understanding of the roles of regulatory mechanisms involving phosphorylation and gene expression in responses to drought stress. The authors delineate both ABA-dependent and ABA-independent regulatory systems and demonstrate crosstalk between these systems.

The review is comprehensive of current developments as well as providing sufficient background of the past twenty years of research in this field. The authors are well established and leading researcher in this field.

Apart from some suggestions for some minor changes to English expression, the manuscript is most suitable for publication in Plants. I have included some suggested changes in the attached document.

>ANSWER

Thank you very much for your constructive and kind suggestions. We corrected the English expression according to the reviewer’s suggestions. In addition, we proofed our revised manuscript with an English editing service.

Reviewer 3 Report

It is an interesting and well organized review article. I have some suggestions to further improve the current version:

Suggestions:

Introduction: why this review is needed? justification needs to be improved in introduction section.

After introduction: Add a small section about plant's response to drought stress.

In line "In this review, we summarizerecentknowledge of how plant " correct to plants

Figure 1, change the font type and size for better readership.

Before conclusion section, authors are suggested to add one new section about the ABA crosstalk with other hormones under drought stress.

General comment: update with recent literature on drought.

Improve language of whole MS and screen for typo errors. Due to no line number, I cannot specify each error.

Author Response

It is an interesting and well organized review article. I have some suggestions to further improve the current version:

Suggestions:

Introduction: why this review is needed? justification needs to be improved in introduction section.

>ANSWER

We deleted several sentences and revised the introduction to focus on protein kinases and transcription factors in the ABA-dependent and ABA-independent pathways.

After introduction: Add a small section about plant's response to drought stress.

>ANSWER

We added new chapter entitled “2. Plant responses to drought stress” to summarize the drought stress responses of plants and explain the relationship between ABA-dependent and ABA-independent pathway under drought stress.

In line "In this review, we summarize recent knowledge of how plant " correct to plants

>ANSWER

We corrected the text according to the suggestions of the reviewer.

Figure 1, change the font type and size for better readership.

>ANSWER

The resolution of our previous Figure 1 was low. We revised the illustration of Figure 1 in this revised manuscript.

Before conclusion section, authors are suggested to add one new section about the ABA crosstalk with other hormones under drought stress.

>ANSWER

We added new section of “7. ABA crosstalk with other hormones under drought stress” in accordance with the suggestion.

General comment: update with recent literature on drought.

>ANSWER

We checked we added recent original papers and reviews in this revised manuscript.

Improve language of whole MS and screen for typo errors. Due to no line number, I cannot specify each error.

>ANSWER

We proofed our revised manuscript with an English editing service.

Round 2

Reviewer 1 Report

The revised version of the manuscript is much better understandable and less redundant.

There are only two points I suggest to change:

Please delete redundancy in lines 52-58.

Please delete the second subheading - plant responses to drought stress and make it a part of the introduction.

Author Response

The revised version of the manuscript is much better understandable and less redundant.

There are only two points I suggest to change:

Please delete redundancy in lines 52-58.

>ANSWER

We deleted and summarized the redundant explanations of ABA-dependent and ABA-independent pathways in the second half of the introduction.

Please delete the second subheading - plant responses to drought stress and make it a part of the introduction.

>ANSWER

Thank you very much for your constructive and kind suggestions. We moved and modified the general explanation of the drought stress responses to the beginning of the introduction.

Reviewer 3 Report

Authors have addressed all of my queries and I have no further comments on the revised version.

Author Response

Authors have addressed all of my queries and I have no further comments on the revised version.

>ANSWER

Thank you very much for your constructive and kind suggestions for our manuscript.